# Lithium crystallization at solid interfaces

Menghao Yang[1], Yunsheng Liu[1] & Yifei Mo [1,2] ✉

Understanding the electrochemical deposition of metal anodes is critical for high-energy rechargeable batteries, among which solid-state lithium metal batteries have attracted extensive interest. A long-standing open question is how electrochemically deposited lithium-ions at the interfaces with the solid-electrolytes crystalize into lithium metal. Here, using large-scale molecular dynamics simulations, we study and reveal the atomistic pathways and energy barriers of lithium crystallization at the solid interfaces. In contrast to the conventional understanding, lithium crystallization takes multi-step pathways mediated by interfacial lithium atoms with disordered and random-closed-packed configurations as intermediate steps, which give rise to the energy barrier of crystallization. This understanding of multi-step crystallization pathways extends the applicability of Ostwald's step rule to interfacial atom states, and enables a rational strategy for lower-barrier crystallization by promoting favorable interfacial atom states as intermediate steps through interfacial engineering. Our findings open rationally guided avenues of interfacial engineering for facilitating the crystallization in metal electrodes for solid-state batteries and can be generally applicable for fast crystal growth.

Crystallization is an important phenomenon in materials science, physics, and chemistry[1–3]. While crystallization induced by the change of temperature or solution is commonly studied, the crystallization under electrochemical deposition remains less explored, despite being a key process in the operation of metal electrodes, such as Li, Na, Mg, and Zn metal anodes for next-generation high-energy rechargeable batteries[4–6]. During electrochemical deposition, metal ions in the electrolyte are deposited and crystalized into metal particles[7–11]. The energy barrier of the crystallization is a key contributor to the overpotential of electrochemical deposition, which should be minimized to improve the electrochemical performance of the metal anode[11]. High overpotential or polarization leads to low power density, reduced materials utilization, low energy efficiency, and even battery failure, such as dendrite growth and short circuiting during the plating of metal electrodes[11–15]. Further improvement of these metal anodes, such as Li metal anode, requires an understanding of crystallization processes during electrochemical metal deposition, especially at the atomistic level.

Using solid electrolyte (SE) to resolve the problems currently plaguing metal anodes is a promising direction, among which solid-state Li metal batteries have attracted great interest[4–6,11–15]. The electrochemical deposition behavior of metal anodes paired with the SEs is distinct from those with liquid electrolytes, as shown in lithium and other metal anodes[4–13]. In liquid electrolytes, the formation and growth of metal particle nuclei during metal plating can be described by the classical nucleation theory[16–19]. By contrast, during the continuous deposition of Li with SEs, the Li-ions transfer across the SE interface, but the subsequent atomistic pathways of how these deposited Li-ions become crystalline Li metal are still not clear. This crystallization process has an intrinsic barrier and is strongly rate-limiting for electrochemical metal plating, as shown in Ag plating with $Ag_4RbI_5$ SE[11,20,21]. For Li and many other metal anodes paired with SEs, the atomistic pathways and kinetic barriers of crystallization are yet to be quantified.

Studying the crystallization processes during electrochemical plating is challenging, owing to the difficulty of directly probing the fast dynamics of individual atoms at the buried SE interfaces. Significant understanding of the crystallization mechanisms has been achieved in colloid systems, which can be visualized at single-particle level[22–25], and in Li metal anodes with liquid electrolyte by cryogenic scanning transmission electron microscopy[16,17]. However, the processes and mechanisms of the crystallization at the SE interface remain

[1]Department of Materials Science and Engineering, University of Maryland, College Park, MD, USA. [2]Maryland Energy Innovation Institute, University of Maryland, College Park, MD, USA. ✉e-mail: yfmo@umd.edu

elusive. Atomistic modeling has unique advantages in directly simulating the atomistic processes at the buried interfaces with real-time resolution (as short as femtosecond $10^{-15}$ s) of every single atom and local energy landscape. In this study, using Li metal anode at the solid interfaces as model systems, we perform large-scale molecular dynamics (MD) simulations to directly reveal the atomistic pathways and energy barriers of crystallization during the plating at the SE interfaces.

## Results

### Pathways of Li metal crystallization

Our atomistic model of Li−SE interface consists of a Li metal slab with (001) surface in contact with (001) surface of $Li_2O$, which is a common interphase layer formed by the reduction of oxides SEs with Li metal[26] (Fig. 1a). The details of the model and the interatomic potentials are described in Methods. To simulate the Li deposition, the Li atoms are randomly inserted crossing the diffusion channels of $Li_2O$ (Methods, Fig. 1a) at the rate of one Li every 2 ps corresponding to a current density of 0.16 nA/nm². By directly modeling the dynamical process of Li insertion with full atomistic details and femtosecond time resolution

(Fig. 1b and Supplementary Fig. 1), the large-scale MD simulations reveal the interface structures and the Li diffusion mechanisms at the Li−SE interfaces[27,28].

An interfacial amorphous lithium layer is formed at the Li−SE interface as a result of the large lattice mismatch between Li metal and the SE[27,28]. In the interfacial amorphous layer, the Li atoms do not have body-centered cubic (BCC) configurations as in the bulk crystalline BCC phase, but instead exhibit local configurations of random hexagonal close-packed (rHCP) Li (which is a random mixture of HCP and face-centered cubic (FCC) stacking) or disordered-Li (which cannot match any known structure prototypes) (Fig. 1a)[28]. For the first Li layer in contact with the SE (2.2 Å from the SE interface in Fig. 2a), most Li atoms are identified as disordered-Li (Fig. 2a and Supplementary Fig. 3). Further from the Li−SE interface, the second to the fourth Li layers (2.2–7.45 Å from the SE interface) contain more rHCP-Li, especially in the fourth Li layer (5.70–7.45 Å) in the vicinity of the bulk BCC crystalline Li. At the fifth Li layer (7.45–9.20 Å from the SE interface) and beyond, most Li atoms are crystalline BCC-Li.

The interfacial atomistic structures at the Li−SE interface play a critical role in the crystallization of metallic lithium during Li

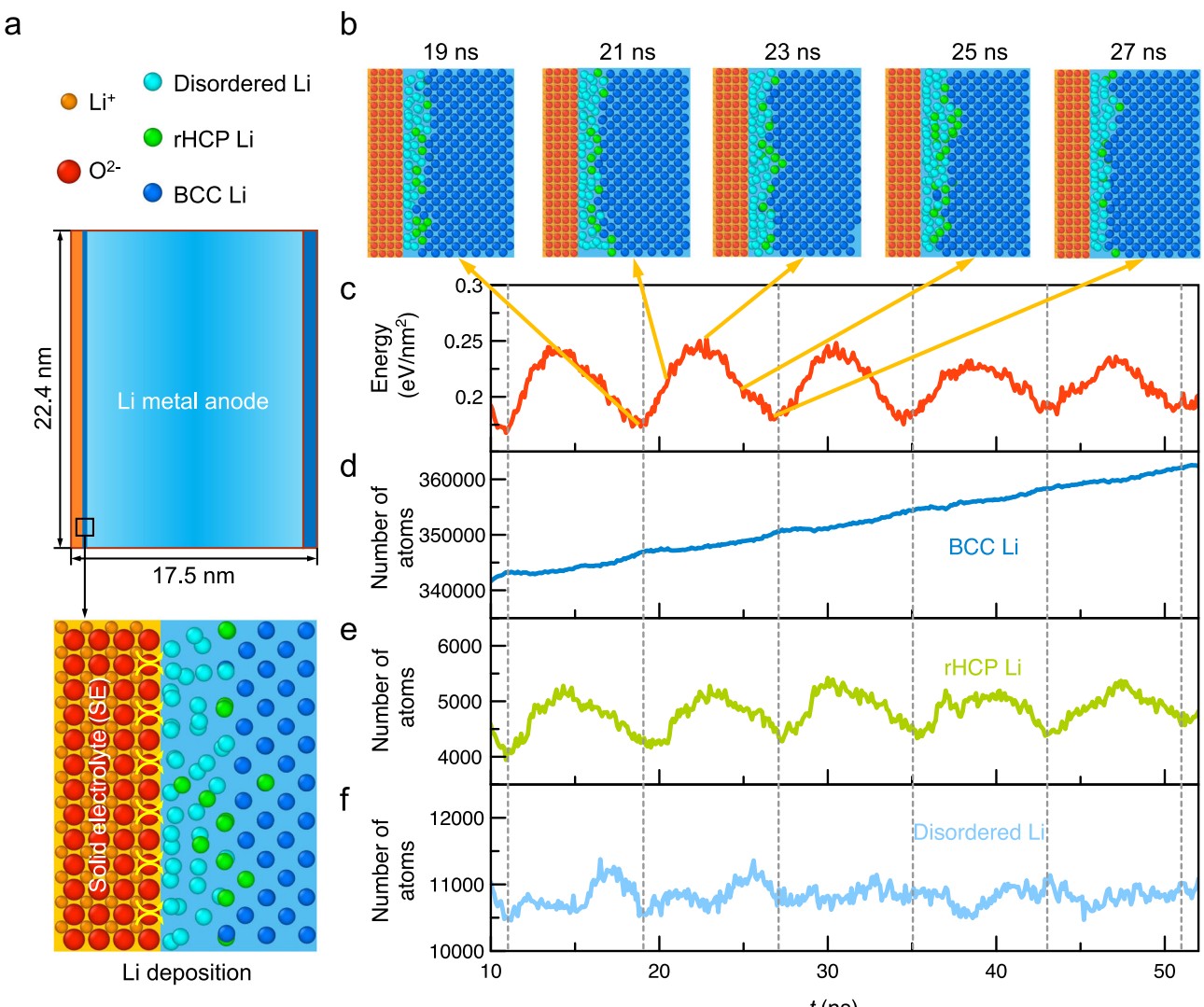

**Fig. 1 | Atomistic modeling of lithium crystallization at solid-electrolyte interface during Li deposition. a** The atomistic model comprises the Li metal slab (light blue) with the solid electrolyte (orange) in the MD simulations. **b** The atomistic structures of the Li−SE interface over a period of energy change during Li deposition. Over the duration of Li deposition, **c** the energy of Li metal slab

referenced to crystalline bulk Li per area ("Methods", Source data are provided as a Source Data file) and **d**–**f** the number of Li atoms with different local configurations, such as body-centered cubic (BCC) and random hexagonal close-packed (rHCP), in the Li metal slab.

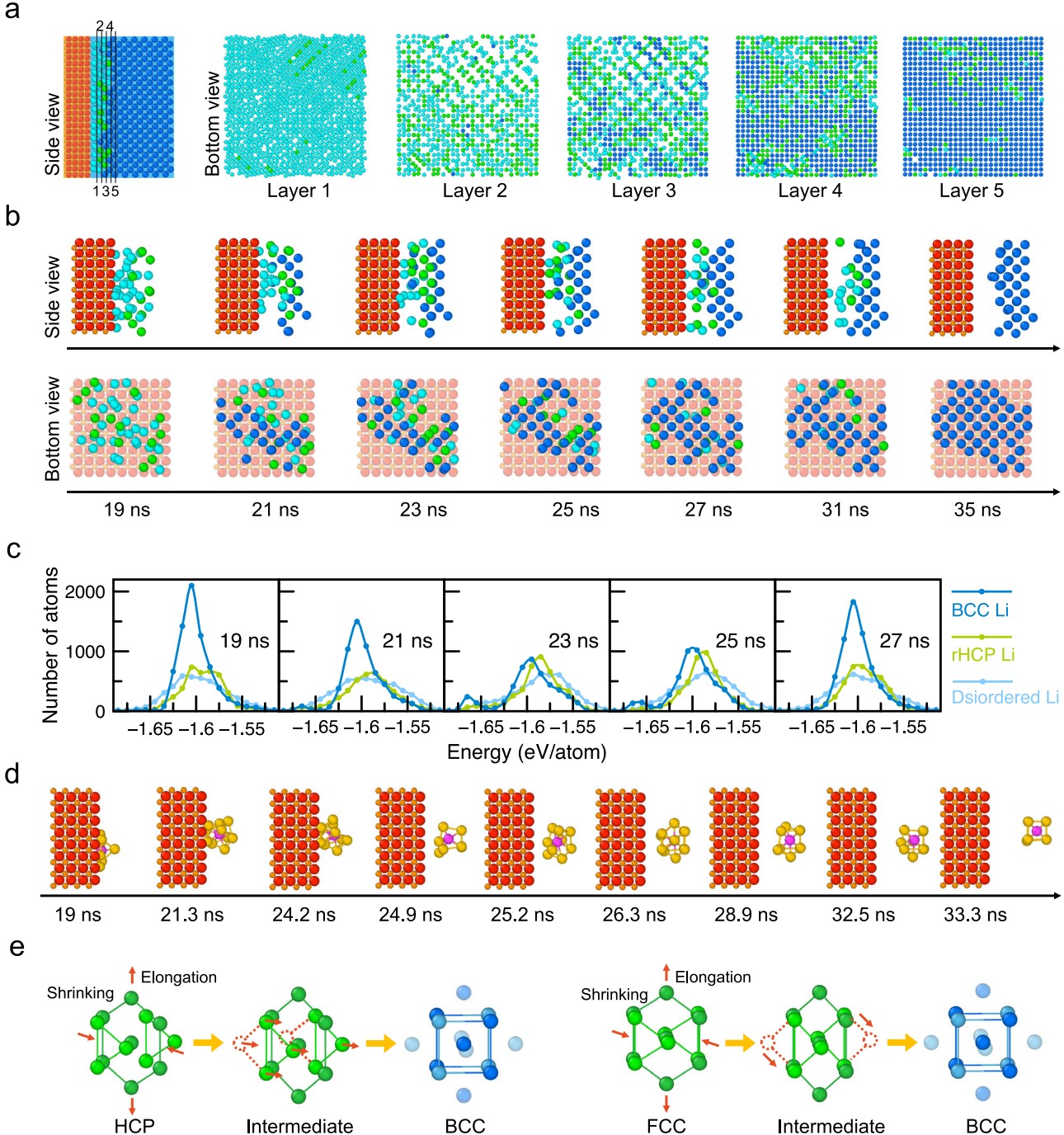

**Fig. 2 | Multiple-step pathway of Li crystallization. a** Interfacial atomistic structures at the Li−SE interfaces at 19 ns, with layer-by-layer bottom view (Disordered-, rHCP (random hexagonal close-packed)-, and BCC (body-centered cubic)-Li are shown in cyan, green, and blue, respectively). The 1st layer is within 2.2 Å from the SE, and each layer beyond is 1.75 Å in thickness. The crystallization process of **b** a group of Li atoms and **d** a single Li atom (purple) and its neighboring Li (yellow). **c** The Li density of atomistic states (DOAS) showing the statistics of the atomistic energies of different Li types (disordered, rHCP and BCC) in the 7.0 Å-thick layer (2nd to 5th layers, 2.2−9.2 Å from the SE). **e** The schematic transition from the HCP (left) or FCC (face-centered cubic, right) configurations to the BCC configuration.

deposition. By tracking the time evolution of Li during MD simulations, we further reveal the atomistic pathways of Li crystallization step-by-step from inserted Li to BCC-Li (Fig. 2b, d and Supplementary Figs. 6, 7). The deposited Li atoms are accommodated by this interfacial amorphous lithium layer (Figs. 1b and 2b) and, as the Li deposition continues, crystalize into BCC-Li metal through two pathways. In one pathway, the deposited Li goes through disordered-Li and then transforms into the crystalline BCC-Li. A major fraction of Li takes another pathway with disordered-Li and goes to the next intermediate, rHCP-Li, before transforming into BCC-Li (Fig. 2b, d).

Therefore, the Li crystallization is mediated by the interfacial amorphous layer at the SE interface, in which the interfacial atoms, disordered-Li and/or rHCP-Li, serves as the intermediates of the multiple-step pathways. These interfacial atoms, i.e. disordered-Li or rHCP-Li, are direct results of the interfacial interactions between SEs and Li metal[27,28].

## Energy barrier of Li crystallization

In order to quantify the energy barrier of Li crystallization, we directly track the energy of the Li metal slab at the SE interface (Fig. 1c),

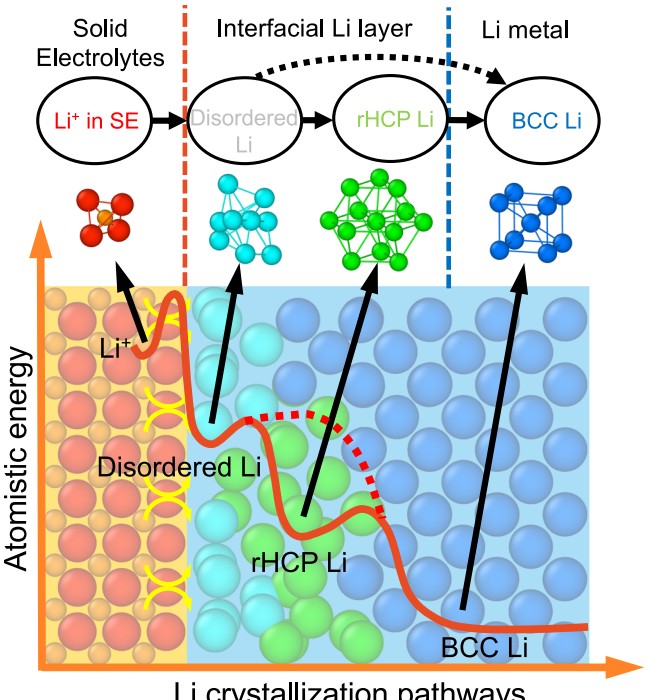

**Fig. 3 | A schematic of multiple-step pathways of Li crystallization.** The Li+ (orange, anion shown in red) in solid electrolytes (SE) goes through disordered-Li (cyan) and/or rHCP (random hexagonal close-packed)-Li (green) in the interfacial Li layer at the SE interface, and transforms into the crystalline BCC (body-centered cubic)-Li metal (blue).

calculated as the energy excess to equilibrium crystalline bulk Li per interfacial area (Methods). This energy fluctuates with a periodicity of 8 ns with peak energies of 0.24–0.25 eV/nm² and energy minima of 0.17–0.18 eV/nm², indicating a total barrier of 0.07–0.08 eV/nm². The periodic energy profile of Li insertion (Fig. 1c) corresponds to the energy barrier of the Li crystallization process, since each period corresponds to the crystallization and growth of a full atomistic layer at the interface of the Li metal slab. The crystallization overpotential, which is evaluated as the potential to insert Li (referenced to the equilibrium potential of crystalline Li bulk) (Methods), ranges from 22 to 38 meV for the five evaluated periods (Supplementary Fig. 2 and Supplementary Note 1) with an average value of 30 meV. The crystallization overpotential, known as an intrinsic barrier and a significant rate-limiting step of electrochemical metal plating[11,20,21], is for the first time quantified for solid-state Li metal deposition.

By tracking the number of disordered- and rHCP-Li during Li deposition (Fig. 1e, f), we find that the trends of rHCP-Li atoms (Fig. 1e) correlate with the energy of the Li metal–SE interface (Fig. 1c), indicating the critical role of rHCP-Li in the process and energy of Li crystallization. In addition, this pathway of Li crystallization through intermediate rHCP-Li is energetically favorable than the crystallization pathway from disordered-Li directly to BCC-Li. Disordered-Li and rHCP-Li in general have higher energies than crystalline BCC-Li, as shown by the atomistic energies of different types of Li atoms by the density of atomistic states (DOAS) of Li[29], in the interfacial amorphous Li layer (2.2–9.2 Å from the SE) (Fig. 2c and Supplementary Figs. 4, 5). Therefore, the higher energies of these non-BCC-Li in the interfacial amorphous Li layer give rise to the energy barrier of crystallization. The rHCP-Li atoms on average exhibit lower atomistic energies than disordered-Li at the peak of energy period of the Li–SE interface (e.g., at 23 ns in Fig. 2c and Supplementary Fig. 4). Therefore, in comparison to the pathway through disordered-Li directly to BCC-Li, rHCP-Li is an

energetically favorable intermediate step in the multi-step pathway of Li crystallization (Fig. 3).

In addition, rHCP (a mix of HCP or FCC) Li configurations transform into BCC-Li through small Li-atom movements, as illustrated in Fig. 2e[24]. When an HCP-Li converts to BCC-Li, the {0001} hexagonal plane becomes the {110} plane by shrinking in the <110> direction or by elongating in the <001> direction, and the other atoms parallel to the hexagonal plane move along the <110> direction to form a BCC configuration. An FCC-Li transforms into a BCC-Li in a similar manner with small Li-atom movements (Fig. 2e). Besides its low energies, the easy transition from rHCP-Li to BCC-Li also makes it a kinetically favorable intermediate of the Li crystallization pathway. This atomistic pathway follows Ostwald's step rule that the higher energy but kinetically favored intermediates form before the final stable states (Fig. 3).

Besides Li(100)–Li₂O(100) interface, we observe the similar energy barrier of Li crystallization and the multiple-step crystallization pathways with rHCP-Li intermediates for other Li metal interfaces with Li₇La₃Zr₂O₁₂ (LLZO) garnet SE (Supplementary Figs. 9 and 10) and LiF (Supplementary Figs. 11 and 12). The same conclusion is expected for sulfides SEs, because Li₂S is a common interphase layer formed by the reduction of sulfides SEs with Li metal[26] and also exhibits a lattice mismatch with Li metal (Fig. 1a)[28]. Therefore, the revealed mechanisms are general for Li–SE interfaces with different SE materials.

## Interface engineering to facilitate Li crystallization

To improve the electrochemical performance of Li metal anodes, it's desirable to lower the energy barrier of Li crystallization, which is a key contributor to the overpotential for the electrochemical deposition[11]. The undesired overpotential caused by the kinetic barrier for Li plating at the Li–SE interface can potentially contribute to the nucleation, formation, and growth of lithium dendrite inside the pores or grain boundaries of SEs, and to the failure of the solid-state battery. Therefore, lowering the barrier of Li crystallization at Li–SE interfaces is important to mitigate dendrite formation in solid-state batteries. Based on the understanding of the multi-step pathways with interfacial atomistic states as intermediates, a rational strategy for facilitating crystallization and mitigating the kinetic barrier is to promote the favorable interfacial-atom intermediate, i.e. rHCP-Li, with lower energy and easier transition to the final BCC-Li state (Fig. 3). These interfacial atom states are determined by the Li–SE interface, and can be tailored by interface engineering.

As an interface-engineering strategy, we introduce fixed HCP-Li nanoclusters (each with 13 atoms) evenly distributed (one per 22.92 Å × 22.92 Å, Fig. 4b) across the Li–SE interface ("Methods"). This model interface with HCP-Li nanoclusters shows a significant increase in the number of rHCP-Li atoms (Fig. 4g), which have lower atomistic energies than disordered-Li as shown in Li DOAS (Fig. 4d and Supplementary Figs. 13, 15). The resulting energy barrier of Li crystallization is 0.04–0.05 eV/nm², significantly lower than 0.07–0.08 eV/nm² for the pristine Li–SE interface (Fig. 4e), confirming the effectiveness of the interface-engineering strategy. Similarly, we find that dopants at the Li–SE interface can also facilitate crystallization. In another model Li–SE interface with Na dopants (one per 22.92 Å × 22.92 Å, Fig. 4a, c) ("Methods"), we observe a similar increase in the number of rHCP-Li and a decrease in the energy barrier of Li crystallization to 0.03–0.04 eV/nm² (Fig. 4e and Supplementary Figs. 14, 15). We investigate a few other dopants, such as K and Ca, and find similar effects in lowering the barrier of Li crystallization (Supplementary Fig. 18). These results demonstrate the interface-engineering strategies to facilitate crystallization. Specifically, engineering the interface tailors the interfacial states of atoms, and promoting the favorable interfacial-atom states as intermediates lowers the barrier of the crystallization. This strategy can serve as a general avenue for improving electrochemical metal plating.

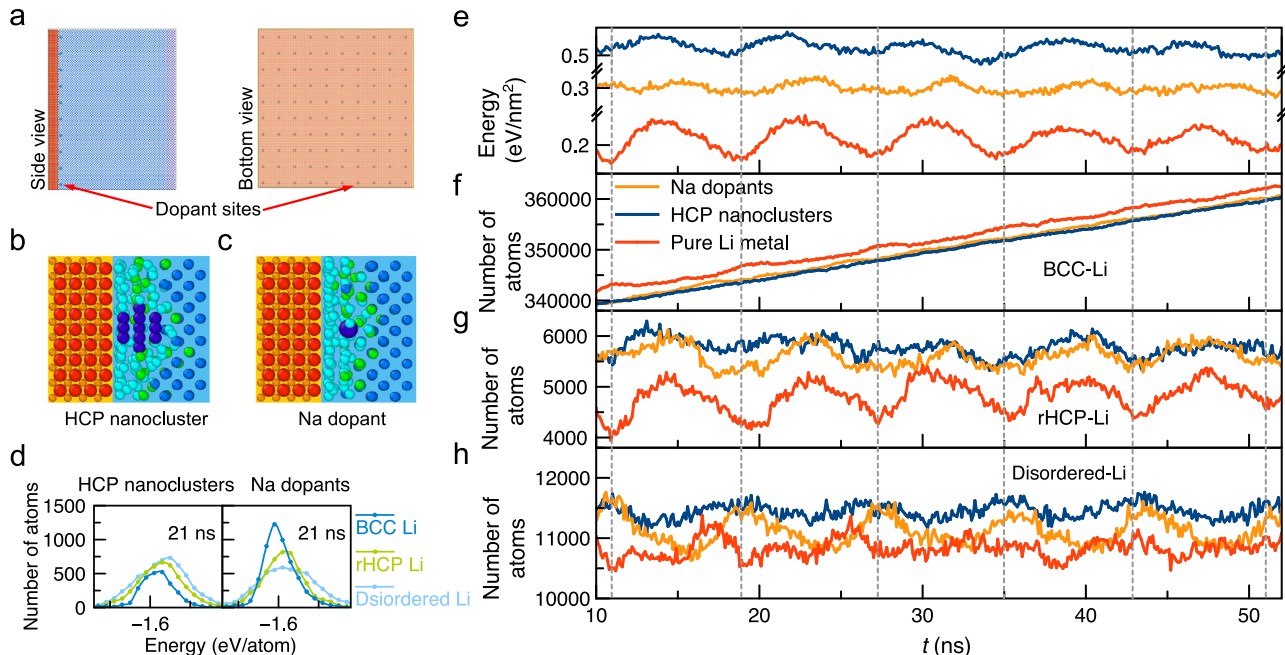

**Fig. 4 | Li crystallization at engineered Li–SE interfaces. a** The Li–SE interface model with nanoclusters or dopants. Atomistic structures of the Li–SE interface with **b** HCP-Li nanoclusters and **c** Na dopants (dark blue), and **d** the atomistic energies of different Li types within the 7.0 Å-thick layer (2nd to 5th layers) shown in Li DOAS. **e** The energy of Li metal referenced to bulk crystalline Li per area (Source data are provided as a Source Data file) and **f–h** the number of Li atoms with different local configurations in the Li metal, during the Li insertion with pristine Li–SE interface (red), interface with Na dopants (orange), and interface with HCP-Li nanoclusters (blue).

## Discussion

The multi-step atomistic pathway of Li crystallization at SE interface unraveled by our MD simulations suggests that Ostwald's step rule has extended applicability to individual atom states. Ostwald's step rule suggests that during the crystallization the higher energy intermediate phases first form before the thermodynamic stable phase. In the multi-step atomistic pathway of Li crystallization (Fig. 3), the higher energy interfacial atom states (e.g., disordered-Li and/or rHCP-Li) are formed first as the intermediate, following the Ostwald's step rule, and then transition to the bulk-phase crystalline atoms (i.e. BCC-Li). The dynamics and energetics of these interfacial atom states during this complex multi-step crystallization process can be elucidated by the density of atomistic states (DOAS)[29] of these interface atoms. The interfacial atom states, which serve as intermediates in the crystallization pathways, are direct results of the interfacial interactions between Li metal and SE, and thus can be tuned by interface engineering. By contrast, in liquid electrolytes, the crystallization is mediated by the surface of the nucleus particles and surface atoms, such as adatoms or vacancies on surfaces including terraces and kinks as illustrated in the Terrace–Ledge–Kink model[7–11,23,24]. This understanding of the multi-step crystallization pathways from the perspective of interfacial atom states leads to our rational strategy for facilitating crystallization through SE interface engineering, as we demonstrated in engineered Li–SE interfaces. These interface-engineering strategies of tuning the atomistic pathways of the crystallization open rationally guided avenues for improving the performance of the electrochemical deposition of metal anodes for high-energy solid-state metal batteries. More generally, similar strategies of tuning interfacial atoms also provide new opportunities for facilitating crystallization in other applications, such as crystal growth.

## Methods

### Li–SE interface model

MD simulations were performed by using large-scale atomic/molecular parallel simulator (LAMMPS) packages[30]. The Li–solid electrolyte (SE) interface model had a dimension of 22.4 nm × 22.4 nm × 17.5 nm and was consisting of Li metal slab with (100) surface in contact with a fixed $Li_2O$ SE (10.5 Å in thickness) with O–terminating (100) facet and a rigid piston (10.5 Å) on the other side of Li metal. The Li(100)–$Li_2O$(100) interface was consisting of 22.4 nm by 22.4 nm (64 × 64 Li unit cells and 49 × 49 $Li_2O$ unit cells), which gave a lattice mismatch of 31%. Periodic boundary conditions were applied upon both directions perpendicular to the interface plane.

### Interatomic potential

For Li metal, the Li interatomic potential was from Nichol et al.[31], which accurately reproduced a variety of properties of Li metal (Supplementary Table 1). To describe the interatomic interactions between Li metal and $Li_2O$, a combination of short-range repulsion and long-range attractions were employed[27,28]. The short-range repulsion between $Li^+$ ion $i$ of $Li_2O$ SE and Li atom $j$ of Li metal with a distance $r_{ij}$ was evaluated as:

$$V_{Li^+-Li}\left(r_{ij}\right) = A_{ij}\exp\left(-\frac{r_{ij}}{\rho}\right),\tag{1}$$

where the values of $A_{ij}$ and $\rho$ were from ref. 32. For LLZO, the values of $A_{ij}$ for $La^{3+}$–Li and $Zr^{4+}$–Li repulsions are set to three and four times that of $Li^+$–Li repulsion interactions, respectively (Supplementary Table 2). The interaction between $O^{2-}$ ion of $Li_2O$ and Li atom $j$ of Li metal was evaluated as:

$$V_{O^{2-}-Li}\left(r_{ij}\right) = \epsilon\left[3\left(\frac{r_m}{r_{ij}}\right)^8 - 4\left(\frac{r_m}{r_{ij}}\right)^6\right],\tag{2}$$

where the values of $\epsilon$ and $r_m$ were obtained from ref. 27. and a cutoff of 10 Å was applied for the power-6 term of long-range attraction. For LLZO, the value of $r_m$ was tuned to fit the Li–O bond length of LLZO, and $\epsilon$ was tuned to fit the interfacial adhesion with Li metal. The

interaction between $F^-$ ion of LiF and Li atom were based on the same formula using the parameters from ref. 33.

For Li(100)–$Li_2O$(100) interface, the interfacial adhesion was calculated as the work of separation of two surfaces to be 0.77 $J/m^2$, which agreed well with the interfacial adhesion of 0.72 $J/m^2$ for Li(100)–$Li_2O$(111) interface[34]. For Li(100)–LLZO(100) interface, the interfacial adhesion was calculated to be 0.77 $J/m^2$, which agreed well with the interfacial adhesion of 0.67–0.98 $J/m^2$ of Li-LLZO interfaces from DFT calculations[35,36]. For the Li(100)–LiF(100) interface, the interfacial adhesion was calculated to be 0.23 $J/m^2$, in comparison to 0.1 $J/m^2$ from DFT calculations[37] (Supplementary Fig. 8 and Supplementary Note 2).

## MD simulations
A timestep of 2 fs was used for MD simulations. The initial model was heated up in NVE with fixed temperature controlled by velocity scaling and a step-by-step increase from 30 to 300 K with an interval of 30 K every 4 ps, and equilibrated at 300 K for 4 ps.

## Lithium deposition
The MD simulations of lithium deposition were performed in the NVE ensemble at 300 K. Li atoms were inserted into the Li insertions sites at the Li–SE interface, which were defined as the points at the centers of four oxygen ions on the top oxygen layer of $Li_2O$ SE, and the deposited Li atoms would subsequently migrate into Li metal slab. During the Li deposition process, one Li atom was inserted every 2 ps into a randomly selected Li insertion site. This insertion rate corresponded to a current density of 0.16 $nA/nm^2$ within the 22.4 nm × 22.4 nm area. The MD simulation of Li plating was performed for over 50 ns, and all analyses were conducted after the initial 10 ns of equilibration.

## Analyses
The plotted values of atomistic energies and coordinates were averaged over 100 ps (50,000 configurations), in order to eliminate the noises caused by fs-level thermal fluctuations during MD simulations. Atomistic configurations were visualized by Ovito software[38], and the local structural environments of Li atoms were classified by the polyhedral template matching (PTM) method[39] with a root-mean-square deviation (RMSD) cutoff of 0.2.

Since the energy $E(t)$ of the Li metal slab was dependent on the number of Li atoms $N(t)$ at time $t$, we plotted (Figs. 1 and 4) the energy of Li metal referenced to crystalline bulk Li $E_{bulk}$ per interface area $A$ defined as follows,

$$\triangle E(t) = \frac{E(t) - E_{bulk} \times N(t)}{A}, \tag{3}$$

where $E_{bulk}$ was the average per-atom energy of perfect crystalline bulk Li metal obtained from MD simulations at 300 K.

The potential to insert Li was evaluated as follows. A total number of $N(t)$ Li atoms was inserted at time $t$. To insert $N(t+\Delta t) - N(t)$ Li atoms during a short time interval $\Delta t$, the potential (referenced to bulk Li metal) was evaluated as

$$\phi_{\triangle t}(t) = \frac{E(t + \triangle t) - E(t)}{N(t + \triangle t) - N(t)} - E_{bulk} \tag{4}$$

For a short time interval $\Delta t$, this potential $\phi_{\triangle t}(t)$ can be understood as instantaneous potential to insert Li at time $t$, as shown in Supplementary Figs. 2b and 16 for $\Delta t$ = 0.5 ns. The average potential of Li insertion from initial time $t_0$ to time $t$ was evaluated as

$$\phi_{t_0}(t) = \frac{E(t) - E(t_0)}{N(t) - N(t_0)} - E_{bulk} \tag{5}$$

This average potential $\phi_{t_0}(t)$ was equivalent to the averaged potential $\phi_{\triangle t}$ over the time period from time $t_0$ to time $t$. In Supplementary Figs. 2c and 17, the averaged potential $\phi_{t_0}(t)$ were plotted with $t_0$ set to the bottom of each energy period in Fig. 1. Since the equilibrium potential was Li bulk, these potentials were equivalent to the overpotential during Li deposition.

## Models of engineered interfaces
For the Li–SE interface with HCP-Li nanoclusters, the HCP nanoclusters consisted of 3-atom top and bottom planes and a middle plane of 7-atom hexagon (Fig. 4b), and were placed with the center of the bottom three atoms 1.75 Å from the top oxygen ions of $Li_2O$ SE (Fig. 4b), and a total of 100 HCP nanoclusters were evenly distributed (one per every 22.92 Å × 22.92 Å, Fig. 4b) at the Li–SE interface. These HCP nanoclusters were fixed with the lattice parameters of $a$ = 3.11 Å and $c$ = 5.09 Å, and the same Li interatomic potential was used to describe the interactions between HCP-Li nanoclusters and Li metal.

For the Li–SE interface with dopants, the dopants were placed at the same positions of the center atom of the HCP nanoclusters with the same distribution (one per 22.92 Å × 22.92 Å) at the Li–SE interface. The interactions between Li and dopants $M$ ($M$ = Na, K, and Ca) were described by Lennard-Jones potential (Supplementary Fig. 18 and Supplementary Note 3). For each Li-$M$ pairs, the parameters $\sigma$ and $\varepsilon$ (Supplementary Table 3) were obtained from the Lorentz–Berthelot rule, respectively, using the values from ref. 40. These simple potentials for dopant-Li interactions correctly reproduce the Li configurations (Supplementary Table 4) and the Li energies (Supplementary Figs. 19 and 20) near the dopant, which are relevant to the physical process studied here. The dopants were fixed. The MD simulation procedures, including relaxation, heating, and Li insertion, of these interfaces were identical to the pristine Li–SE interfaces.

## Data availability
The data that support the findings of this study are available from the corresponding author upon request. Source data are provided with this paper.

## Code availability
The code used in this study is available from the corresponding author upon request.

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

## Acknowledgements

We acknowledge the support from the Office of Energy Efficiency and Renewable Energy, U.S. Department of Energy, and the National Science Foundation Award # 2004837. We acknowledge the computational facilities from the University of Maryland supercomputing resources and the Maryland Advanced Research Computing Center (MARCC).

## Author contributions

Y.M. conceived and supervised the project. Y.M. and M.Y. designed the computation and analysis, M.Y. performed and analyzed the MD simulations, and Y.L. designed and performed the DFT calculations. Y.M. and M.Y. wrote the manuscript.

## Competing interests

The authors declare no competing interests.
