## [Peer Review File · Nature Communications]

Reviewer comments

Reviewer #1 (Remarks to the Author):

I appreciate the authors' substantial effort at revising the manuscript. While it is certainly improved, there are still major issues that need to be resolved before I can recommend it for publication.

1. The title of the manuscript is unchanged and misleading. As noted, the authors did not simulate a Li-SE interface. A Li₂O model system is not a SE. This needs to be corrected.
2. Li₂O is one of the phases that are formed at the SE/Li interface for oxide SEs. It does not form for sulfide or other SE chemistries. Hence, the statement that Li/Li₂O interfaces are "ubiquitous" is debatable. This is a point that I am willing to concede if the authors carefully clarify the limitations of their study and its scope of applicability to oxide SE chemistries.
3. The authors added simulations of LLZO and LiF with Li metal. However, there are no details on the potentials used for these simulations.
4. I appreciate the authors providing the validation of the potential. While the potential clearly works reasonably well in reproducing the BCC bulk properties, the surface properties (e.g., surface energies) and fcc and hcp properties leave much to be desired. I am also puzzled why the validation on the fcc and hcp properties is limited to just the lattice parameters. Quite clearly, the source of DFT data they used (the Materials Project) contains the elastic constants and surface energies of these phases as well. More critically, a key claim of the paper is the crystallization path takes place through multiple phases. Arguably, it is even more important that the fcc/hcp phases are correctly captured.
5. Besides the validation of the fcc/hcp phases above, another property that is missing is the energy differences between the fcc/hcp and the bcc phase. This would seem to be a critically important number to report, given that any large errors in these values would affect the crystallized phases at the interphase.

This reviewer will note that it is accepted that the simple potentials used in this work cannot reproduce all properties for Li and Li₂O and the interfaces. But at the minimum, the critical validation data that can dramatically affect the conclusions need to be reported and if necessary, the authors should identify potential uncertainty in their conclusions.

Reviewer #3 (Remarks to the Author):

In this manuscript, the authors investigated lithium crystallization at solid electrolyte interface using classical molecular dynamics. They revealed the atomistic pathways and energy barriers of lithium crystallization at the solid-electrolyte interfaces. The results of this study are interesting, but the following concerns should be addressed before its publication.

1. The authors should verify the interatomic potentials for Li₂O, LLZO, and LiF. The authors can compare the calculated physical properties with experimental results for Li₂O, LLZO, and LiF, as they

did for Li metal in Table S1. Also, please provide the details of interatomic potential as a Table for Li₂O, LLZO, and LiF.

2. The authors should provide the details of the termination of Li₂O, LLZO, and LiF slabs, and verify that those slabs exhibit the lowest surface energy.

3. As the SE are fixed in this calculations, dynamic behaviors of SE are ignored. This calculation cannot capture the exact lithium crystallization at the interface.

4. The authors should verify the interatomic potentials for Na, Ca, and K, as they did for Li metal in Table S1.

Response to Reviewers

We thank all reviewers for their helpful comments and suggestions. Following the reviewers' suggestions, we have provided additional information to support our results and claims. We have also significantly revised our manuscript regarding the validation of our methods and highlighted the broader significance of our results. The detailed point-by-point responses to reviewers are included below. We have fully addressed all reviewers' comments, and we greatly appreciate the helpful comments from the reviewers for improving our manuscript.

Response to Reviewer #1.

"I appreciate the authors' substantial effort at revising the manuscript. While it is certainly improved, there are still major issues that need to be resolved before I can recommend it for publication."

We thank the reviewer again for reviewing our manuscript. In this revision, we have addressed the comments and suggestions from the reviewer. We greatly appreciate the helpful comments from the reviewers for improving our manuscript.

"1. The title of the manuscript is unchanged and misleading. As noted, the authors did not simulate a Li-SE interface. A Li₂O model system is not a SE. This needs to be corrected."

Following the reviewer's comment, we revised the title into "Lithium Crystallization at Solid Interfaces". We hope this revised title addressed the concerns of the reviewer. If this title is still not satisfactory, we would be happy to take any specific suggestions from the reviewer. The rationale for the previous title was explained in the previous response, we think this Li-Li₂O interface (and similarly the added Li-LiF interfaces) can be considered as the actually formed interface of Li-SE interfaces, because many SEs are not thermodynamically stable (except for the Li-LLZO interface added). We also revised a number of places in the abstract, introduction, and discussion. We would also revise other places where the reviewer feels necessary.

"2. Li₂O is one of the phases that are formed at the SE/Li interface for oxide SEs. It does not form for sulfide or other SE chemistries. Hence, the statement that Li/Li₂O interfaces are

"ubiquitous" is debatable. This is a point that I am willing to concede if the authors carefully clarify the limitations of their study and its scope of applicability to oxide SE chemistries."

We appreciate the rigor raised by the reviewer. We agree with the reviewer's comment. Therefore, this word is removed and we revised the sentence as follows: "... interface consists of a Li metal slab with (001) surface in contact with (001) surface of Li₂O, which is a common interphase layer formed by the reduction of oxides SEs with Li metal (Fig. 1a)."

The reviewer also raised a very good point about sulfide SEs, which we completely agree as we did not cover sulfide SEs in our study. As revealed in the previous computational and experimental studies by our group and many groups, most sulfide SEs form Li₂S at the Li interface. Given Li₂S has the same crystal structures as Li₂O, our conclusion would apply to these interfaces, as Li₂S also has a lattice mismatch with Li metal and would form a similar Li amorphous layer. We add the following discussion about the limitation and applicability of our study as follows: "The same conclusion is expected for sulfides SEs, because Li₂S is a common interphase layer formed by the reduction of sulfides SEs with Li metal (26) and also exhibits a lattice mismatch with Li metal (Fig. 1a)"

"3. The authors added simulations of LLZO and LiF with Li metal. However, there are no details on the potentials used for these simulations."

We thank the reviewer for the careful review of our manuscript. The potential was adopted from our previous work, in which we have tested the LLZO and LiF with Li metal interface for their interfacial adhesion. The previous work was cited in the corresponding parts of the potential (Ref. 28). We here include these into the Methods and SI: "For LLZO, the values of A_{ij} for La³⁺-Li and Zr⁴⁺-Li repulsions are set to three and four times, respectively, as large as Li⁺-Li repulsion interactions." and "For LLZO, the value of r_m was tuned to fit the Li-O bond length of LLZO, and ϵ was tuned to fit the interfacial adhesion with Li metal. The interaction between F⁻ ion of LiF and Li atom were based on the same formula using the parameters from Ref. (33)" The potential values are added into SI (Table S2) and also referred in the Methods.

Table S2. Short-range potential parameters for Li₂O, LiF and LLZO(100) with Li metal.

Pairs	A_{ij} (eV)	ρ (Å)
Li ₂ O: Li ⁺ -Li	465.54	0.2939
LiF: Li ⁺ -Li	2575.00	0.26
LLZO: La ³⁺ -Li	1396.62	0.2939
LLZO: Zr ⁴⁺ -Li	1862.16	0.2939
	ϵ (eV)	r_m (Å)
Li ₂ O: O ²⁻ -Li	0.273	2.00
LiF: F ⁻ -Li	0.198	2.04
LLZO: O ²⁻ -Li	0.287	1.90

“4. I appreciate the authors providing the validation of the potential. While the potential clearly works reasonably well in reproducing the BCC bulk properties, the surface properties (e.g., surface energies) and fcc and hcp properties leave much to be desired. I am also puzzled why the validation on the fcc and hcp properties is limited to just the lattice parameters. Quite clearly, the source of DFT data they used (the Materials Project) contains the elastic constants and surface energies of these phases as well. More critically, a key claim of the paper is the crystallization path takes place through multiple phases. Arguably, it is even more important that the fcc/hcp phases are correctly captured.”

Here we added the energies, surface energies, and elastic modulus in the updated Table S1 as requested by the reviewer. All these properties predicted by EAM potential show reasonable agreement with DFT. We provide more explanations concerning FCC and HCP phases in the response to the next comment below.

Table S1. Calculated physical properties for bulk lithium phrases. Lattice constants a_0 and c_0 in Å, elastic constants (C_{11} , C_{12} , and C_{44}) and bulk modulus B_0 in GPa, surface energies γ in J/m², cohesive energies E_{coh} and vacancy formation energies E_{vac} in eV/atom, diffusion coefficients D_{Li} in 10⁻¹¹cm²/s, the energy of FCC/HCP phase ($E_{\text{FCC}}/E_{\text{HCP}}$) above the BCC phase in eV/atom.

		Classical potential (3)	DFT (4)	Experimental values
BCC	a_0	3.51	3.43	3.49 (5)
	C_{11}	20	15	13.42 (6)
	C_{12}	12	13	11.30 (6)
	C_{44}	10	11	8.89 (6)
	B_0	14.5	14	13.3 (7), 12.0 (6)
	γ_{100}	0.29	0.46	
	γ_{110}	0.33	0.50	0.522 (8), 0.525 (9)
	γ_{111}	0.41	0.54	
	E_{coh}	1.65	1.61 (10), 1.68 (10)	1.69 (11)
	E_{vac}	0.54	0.53 (12)	
	D_{Li}	3.05	1 – 10 (13)	5 – 9 (14)
FCC	a_0	4.55	4.32	
	E_{FCC}	0.023	-0.002	
	C_{11}	22	-5	
	C_{12}	11	23	
	C_{44}	16	10	
	B_0	18	14	
	γ_{100}	0.38	0.47	
	γ_{110}	0.39	0.54	
	γ_{111}	0.60	0.50	
HCP	a_0	3.22	3.08	3.11 (5)
	c_0	5.26	4.92	5.09 (5)
	E_{HCP}	0.023	-0.002	
	C_{11}	29	22	
	C_{12}	7	11	
	C_{44}	19	6	
	B_0	24	14	
	γ_{0001}	0.70	0.53	
	γ_{10-10}	0.45	0.53	
		γ_{11-20}	0.32	0.51

“5. Besides the validation of the fcc/hcp phases above, another property that is missing is the energy differences between the fcc/hcp and the bcc phase. This would seem to be a critically important number to report, given that any large errors in these values would affect the crystallized phases at the interphase.

This reviewer will note that it is accepted that the simple potentials used in this work cannot reproduce all properties for Li and Li₂O and the interfaces. But at the minimum, the critical validation data that can dramatically affect the conclusions need to be reported and if necessary, the authors should identify potential uncertainty in their conclusions.”

We thank the reviewer for the insightful comments. We in principle agree with the reviewer but would like to note the nuances below. We added the energies of FCC and HCP phases in Table S1 (shown above). In the previous response, we did not include the energies of FCC/HCP, because 1) many DFT databases show different energies for these BCC/FCC/HCP phases and 2) DFT in general show FCC/HCP have lower 0K energies than BCC. For example, in the Materials Project, BCC Li is predicted to be 2 meV/atom higher energy than FCC and HCP Li. In addition, most DFT calculations show the energies of these three phases are within a few meV/atom, which is in general the error range of DFT (Thus, the MP displays all these phases as below <0.01eV/atom). In the literature, the papers comparing these Li phases mostly indicate the difficulty to distinguish the 0 K energies between these Li phases (*Nature* **400**, 141-144 (1999); *J. Appl. Phys.* **123**, 065901 (2018)). In addition, DFT predicted FCC Li has negative elastic modulus C11 at zero pressure, which may also indicate some possible problem of DFT in evaluating these phases at zero pressure. Therefore, given the lack of adequately reliable benchmark values of energies for comparing EMA results, we did not include them in the last round. Here we include these energy values of the FCC and HCP phases in Table S1, so the reviewer and readers can see the results, but we would caution about some properties.

We agree with the reviewer very much that the relevant properties related to the conclusion and mechanisms should be validated. The EAM potential used here correctly reproduces the BCC Li as the stable phase at 300K. In addition, there is good agreement with cohesive and vacancy energies, lattice parameters, elastic modulus, surface energies/interfacial adhesion, and diffusion coefficients, which are all very critical to the physical process studied. In addition,

we would like to point out that the HCP/FCC bulk phase is not entirely identical to the individual atom states of amorphous/rHCP Li in our interphase layer (though some fundamental interactions may be relevant). To test and verify of these rHCP Li and in the presence of dopant, we design the test as follows, which were added into the SI: “**Dopant effect on energies**. We here test the effect of Na, Ca, and K dopants on the energies and configurations of Li atoms. As described in Methods, the potential parameters for Li- M (Na, Ca and K) interactions were listed in Table S3. In the BCC Li metal with a single dopant M , we calculated the Li- M bond length and Li-Li bond length for those nearest-neighbor Li atoms of the dopant. The Li- M and nearby Li-Li bond length under the presence of M dopant show good agreement between DFT and the classical potentials (Table S4).

We conducted the following test to verify the effect of dopant on the energies of Li atoms, which is relevant for the intermediate steps of the crystallization pathway. We constructed supercells of BCC and HCP Li metal with a total of 128 and 108 atoms, respectively. For each supercell, one Li atom was replaced by a Na, Ca and K dopant. To generate a range of atomic configurations mimicking those in the Li-SE interface, we performed MD simulations for 100 ps at 300 K in the NPT ensemble, and selected 20 configuration snapshots (1 ps each) from the last 20 ps for both Li phases. For all these snapshots of atomic configurations with a dopant in BCC and HCP Li, we evaluated and compared the energies predicted by the interatomic potentials and DFT calculations without static relaxations. The energies for every configuration are compared in Fig. S20 and for the distribution in Fig. S19. For most configurations of BCC and HCP Li under Na, Ca, and K dopant, there are strong correlations between the energies of DFT and interatomic potentials (Fig. S20). The distribution of all configurations also shows the effect of the dopant on the energies of BCC and HCP Li agree between interatomic potential and DFT.

Table S4. Comparing Li- M dopant and Li-Li bond length for DFT and classical potential. The percentage number in parentheses shows the deviation from the original Li-Li bond in Li metal.

	Li- M bond length (Å)		Li-Li bond length (Å) near M dopant	
	DFT	potential	DFT	potential
Na	3.22 (8.3 %)	3.33 (9.7 %)	3.71 (8.3 %)	3.85 (9.7 %)
Ca	3.29 (10.6 %)	3.40 (11.7 %)	3.79 (10.6 %)	3.92 (11.7 %)
K	3.42 (15.2 %)	3.71 (22.0 %)	3.95 (15.2 %)	4.23 (20.4 %)

Figure S19. The energy distributions of BCC and HCP Li supercells with a single **a)** Na, **b)** Ca, and **c)** K dopant atom by interatomic potentials, and BCC and HCP Li supercells with a single **d)** Na, **e)** Ca, and **f)** K dopant atom by DFT calculations. All energies are referenced to the lowest energy configuration of the BCC Li with a dopant.

Figure S20. Comparing the energy for every configuration of BCC and HCP Li with a single **a)** Na, **b)** Ca, and **c)** K dopant by interatomic potential and DFT. All energies are referenced to the lowest energy configuration of the BCC Li with a dopant.”

The qualitative agreements between DFT and interatomic potentials validate our potentials on rHCP Li and support the revealed mechanisms and conclusions.

Response to Reviewer #3.

“In this manuscript, the authors investigated lithium crystallization at solid electrolyte interface using classical molecular dynamics. They revealed the atomistic pathways and energy barriers of lithium crystallization at the solid-electrolyte interfaces. The results of this study are interesting, but the following concerns should be addressed before its publication.”

We thank the reviewer for the positive comments about our study.

“1. The authors should verify the interatomic potentials for Li₂O, LLZO, and LiF. The authors can compare the calculated physical properties with experimental results for Li₂O, LLZO, and LiF, as they did for Li metal in Table S1. Also, please provide the details of interatomic potential as a Table for Li₂O, LLZO, and LiF.”

We thank the reviewer for the careful review of our manuscript. The potential was adopted from our previous work, in which we have tested the LLZO and LiF with Li metal interface for their interfacial adhesion. The previous work was cited in the corresponding parts of the potential (Ref. 28). We here include these into the Methods and SI: “For LLZO, the values of A_{ij} for La³⁺-Li and Zr⁴⁺-Li repulsions are set to three and four times, respectively, as large as Li⁺-Li repulsion interactions. ” and “For LLZO, the value of r_m was tuned to fit the Li-O bond length of LLZO, and ϵ was tuned to fit the interfacial adhesion with Li metal. The interaction between F⁻ ion of LiF and Li atom were based on the same formula using the parameters from Ref. (33).” The potential values are added into SI (Table S2).

Table S2. Short-range potential parameters for Li₂O, LiF and LLZO(100) with Li metal.

Pairs	A_{ij} (eV)	ρ (Å)
Li ₂ O: Li ⁺ -Li	465.54	0.2939
LiF: Li ⁺ -Li	2575.00	0.26
LLZO: La ³⁺ -Li	1396.62	0.2939
LLZO: Zr ⁴⁺ -Li	1862.16	0.2939
	ϵ (eV)	r_m (Å)
Li ₂ O: O ²⁻ -Li	0.273	2.00
LiF: F ⁻ -Li	0.198	2.04
LLZO: O ²⁻ -Li	0.287	1.90

The potential is verified for their interfacial adhesion, as we now mentioned in the text: “For the Li(100)–LiF(100) interface, the interfacial adhesion was calculated to be 0.23 J/m², in comparison to 0.1 J/m² from DFT calculation (15). For the Li(100)–LLZO(100) interface, the interfacial adhesion was calculated to be 0.77 J/m², which agreed well with 0.67–0.98 J/m² of Li-LLZO interfaces from DFT calculations (16, 17).” In addition, in our previous work, we also test the Li cycling and plating, showing pore formation in agreement with experiments.[*Angew. Chem. Int. Ed.* 2021, **60**, 21494–21501]

“2. The authors should provide the details of the termination of Li₂O, LLZO, and LiF slabs, and verify that those slabs exhibit the lowest surface energy.”

We thank the reviewer for the comments. All surface terminations we used in our study are the lowest energy surfaces according to the DFT values from the literature (as shown in this table).

	Surface Termination	Surface Energy (J/m ²)
LiF ^[1]	(100)	0.36
	(110)	0.84
LLZO ^[2]	(100) Zr-poor	0.74
	(100) Zr-rich	0.83
	(001) Zr-poor	0.82
	(001) Zr-rich	0.92
	(101) Zr-poor	0.88
	(101) Zr-rich	0.91
	(110) Zr-poor	0.90
	(110) Zr-rich	0.95
Li ₂ O ^[3]	(100) Li-rich	6.28
	(100) O-rich	1.99
	(110) Li-rich	6.62
	(110) O-rich	2.52
	(111) Li-rich	8.12
	(111) O-rich	1.19

[1] *J. Electrochem. Soc.* **163**, A592 (2016)

[2] *ACS Appl. Mater. Interfaces* **12**, 16350-16358 (2020)

[3] *Nanotechnology* **20**, 445703 (2009).

We also show the picture of the atomistic structures of these surface terminations in SI and Fig. S8.

Figure S8. Atomistic structures of the termination for **a)** Li₂O (Li⁺: green, O²⁻: red), **b)** LiF (F⁻: dark green), **c)** LLZO (100) surfaces (La³⁺: red, Zr⁴⁺: orange, O²⁻: light green). All surface terminations are stoichiometric and are constructed by cutting at the face of the unit cell.

In our model, the SE provides a fixed medium for Li deposition, and for the interfacial interaction of the Li-SE interface. Thus the interactions of these SEs with Li are important. However, given the SE is fixed, no potential is needed for within the SEs. Given the important role of interfacial adhesion (rather than the SE surface to vacuum) in determining the interface behavior and mechanisms as shown in previous work (*Adv. Mater.* **33**, 2008081 (2021); *Chem. Rev.* **120**, 7745-7794 (2020)), we have validated the interface adhesion of Li-SE interface, which was calculated as the work of separation of two surfaces: “For the Li(100)–LiF(100) interface, the interfacial adhesion was calculated to be 0.23 J/m², in comparison to 0.1 J/m² from DFT calculations (3). For the Li(100)–LLZO(100) interface, the interfacial adhesion was calculated to be 0.77 J/m², which agreed well with 0.67–0.98 J/m² of Li-LLZO interfaces from DFT calculations (4, 5). ”

“3. As the SE are fixed in this calculations, dynamic behaviors of SE are ignored. This calculation cannot capture the exact lithium crystallization at the interface.”

We thank the reviewer for the comments. The SEs have much higher modulus than Li metal (as shown in Table R2). Therefore, the deformation of SE is insignificant in relation to the change of Li at the interface. In addition, these studied SE materials are chemically stable with Li, so there would be no decomposition of the SEs. In our model, the SE only offers the medium for Li-ion deposition channels and interfacial adhesion. Given we focus on the

dynamics of the Li metal at the interface, this assumption of SE would not change our results. The dynamic changes are the interphase layer of Li, which shows a unique mechanism of crystallization. The SEs would have little change during the dynamic changes of interfacial lithium metal due to their much higher modulus. For these reasons, fixing SEs would not qualitatively change our conclusion or the revealed mechanisms.

Table R2. Elastic properties of Li metal and SE from experiments and calculations.

	Li metal ^[1]	Li ₂ O ^[2,3]	LiF ^[4]	t-Li ₇ La ₃ Zr ₂ O ₁₂ ^[5]
Young's modulus (GPa)	2.83	162.9	125.67	149.8
Shear modulus (GPa)	7.82	69.8	52.35	59.6
Poisson's ratio	0.381	0.17	0.20	0.26

[1] *J. Mater. Sci.* **54**, 2585-2600 (2019).

[2] *J. Electrochem. Soc.* **163**, A67-A74 (2016).

[3] *J. Nucl. Mater.* **160**, 125 (1988).

[4] *Mater. Chem. Phys.* **244**, 122733 (2020).

[5] *J. Mater. Sci.* **47**, 5970 (2012).

“4. The authors should verify the interatomic potentials for Na, Ca, and K, as they did for Li metal in Table S1.”

We thank the reviewer for the comment. We agree with the reviewer that one should validate whether the potential can correctly reproduce the process or properties related to the studied mechanisms. In our case, the key question is how these dopants affect nearby Li configurations and their energies, so our conclusions on the mechanisms of Li crystallization are correct. We verified their relevant properties, such as the change of Li configuration and energies as added into SI and explained below. For the dopants, we adopt the scheme proposed by Pound *et. al.* (*Phys. Stat. Solidi* **30**, 619-623 (1975)) for the interactions between Li metal and dopants *M*, and Li-*M* bond length and defect formation energy have been validated for these dopants. To test and verify of these rHCP Li and in the presence of dopant, we design the test as follows, which were added into the SI: **“Dopant effect on energies.** We here test the effect of Na, Ca, and K dopants on the energies and configurations of Li atoms. As described in Methods, the potential parameters for Li-*M* (Na, Ca and K) interactions were listed in Table S3. In the BCC Li metal with a single dopant *M*, we calculated the Li-*M* bond length and Li-

Li bond length for those nearest-neighbor Li atoms of the dopant. The Li-M and nearby Li-Li bond length under the presence of M dopant show good agreement between DFT and the classical potentials (Table S4).

We conducted the following test to verify the effect of dopant on the energies of Li atoms, which is relevant for the intermediate steps of the crystallization pathway. We constructed supercells of BCC and HCP Li metal with a total of 128 and 108 atoms, respectively. For each supercell, one Li atom was replaced by a Na, Ca and K dopant. To generate a range of atomic configurations mimicking those in the Li-SE interface, we performed MD simulations for 100 ps at 300 K in the NPT ensemble, and selected 20 configuration snapshots (1 ps each) from the last 20 ps for both Li phases. For all these snapshots of atomic configurations with a dopant in BCC and HCP Li, we evaluated and compared the energies predicted by the interatomic potentials and DFT calculations without static relaxations. The energies for every configuration are compared in Fig. S20 and for the distribution in Fig. S19. For most configurations of BCC and HCP Li under Na, Ca, and K dopant, there are strong correlations between the energies of DFT and interatomic potentials (Fig. S20). The distribution of all configurations also shows the effect of the dopant on the energies of BCC and HCP Li agree between interatomic potential and DFT.

Table S4. Comparing Li-M dopant and Li-Li bond length for DFT and classical potential. The percentage number in parentheses shows the deviation from the original Li-Li bond in Li metal.

	Li-M bond length (Å)		Li-Li bond length (Å) near M dopant	
	DFT	potential	DFT	potential
Na	3.22 (8.3 %)	3.33 (9.7 %)	3.71 (8.3 %)	3.85 (9.7 %)
Ca	3.29 (10.6 %)	3.40 (11.7 %)	3.79 (10.6 %)	3.92 (11.7 %)
K	3.42 (15.2 %)	3.71 (22.0 %)	3.95 (15.2 %)	4.23 (20.4 %)

Figure S19. The energy distributions of BCC and HCP Li supercells with a single **a)** Na, **b)** Ca, and **c)** K dopant atom by interatomic potentials, and BCC and HCP Li supercells with a single **d)** Na, **e)** Ca, and **f)** K dopant atom by DFT calculations. All energies are referenced to the lowest energy configuration of the BCC Li with a dopant.

Figure S20. Comparing the energy for every configuration of BCC and HCP Li with a single **a)** Na, **b)** Ca, and **c)** K dopant by interatomic potential and DFT. All energies are referenced to the lowest energy configuration of the BCC Li with a dopant.

”

The qualitative agreements between DFT and interatomic potentials validate our potentials on rHCP Li and support the revealed mechanisms and conclusions.

Again, we thank all reviewers for carefully reading the manuscript and for their valuable comments and suggestions. We have addressed all of them in this response. We greatly appreciate the high standard held by the reviewers.

Reviewer comments further

Reviewer #1 (Remarks to the Author):

I thank the authors for the comprehensive response. I can recommend the manuscript for publication as is. I believe the insights are useful, even if I still have concerns about the reliability of the potential used to obtain those insights.

As a side comment (a response from the authors is not needed for publication), the authors should seriously consider fitting a custom interatomic potential to their target system of interest in future works. The main weakness of this work stems from concerns about the interatomic potential used. For studies of a reactive interface between Li and Li₂O/LiF, it is critical that the energetics are captured properly and quite simply, one cannot expect that of a traditional interatomic potential. Machine learning interatomic potentials are by now a very well-established field and application to "simple" systems such as Li/Li₂O or Li/LiF would be relatively straightforward with many tools publicly available for the purpose.

Reviewer #3 (Remarks to the Author):

The authors have addressed my comments and suggestions. Overall, I recommend acceptance of this revised version.

Response to Reviewer #1.

“I thank the authors for the comprehensive response. I can recommend the manuscript for publication as is. I believe the insights are useful, even if I still have concerns about the reliability of the potential used to obtain those insights.

As a side comment (a response from the authors is not needed for publication), the authors should seriously consider fitting a custom interatomic potential to their target system of interest in future works. The main weakness of this work stems from concerns about the interatomic potential used. For studies of a reactive interface between Li and Li₂O/LiF, it is critical that the energetics are captured properly and quite simply, one cannot expect that of a traditional interatomic potential. Machine learning interatomic potentials are by now a very well-established field and application to "simple" systems such as Li/Li₂O or Li/LiF would be relatively straightforward with many tools publicly available for the purpose.”

We thank the reviewer again for reviewing our manuscript, for careful checking, and for helpful suggestions. We are aware of the rapid progress in the field of machine learning interatomic potentials (MLP) and many well-established, easy-to-use software packages for them. We recently have a few recent publications on this topic (e.g. Ref 29. S. Wang, Y. Liu, Y. Mo, Frustration in Super-Ionic Conductors Unraveled by the Density of Atomistic States. Angew. Chem. Int. Ed., e202215544 (2023).)

We indeed seriously considered the use of MLPs for this project. We agree with the reviewer that for a simple interface of Li/Li₂O or Li/LiF, it would not be difficult, with some proper consideration of van de Waals or long-range interactions. What we find more problematic is the issues related to Li metal as the reviewers referred to in the previous responses. One big problem is that DFT in general tends to over-stabilize FCC/HCP phases over BCC phases of Li. This issue may be tricky to fix. Another issue is that some tricky errors can easily happen during MD simulations using MLPs. For our large-scale MD simulations over a long duration, this can cause the failure of the simulation or some unphysical phenomena. This issue is also very difficult to fix, as conventional training and testing of MLPs focus on average force/energy errors or materials properties and do not directly address these. We have a separate paper to discuss this issue. These are the reasons that we did not use MLPs in this

study.

In general, we agree that properly optimized and tested MLPs should achieve a better physical description, in particular, for those dopants at interfaces, which may involve some more complex interactions. We also agree that interface charge transfer implied by the reviewer may be handled better. Nonetheless, we believe our results provide interesting insights into this very important topic and motivate future studies, many of which may hopefully be based on MLPs.